# PRIVACY-PRESERVING REPRESENTATION LEARNING BY DISENTANGLEMENT

## ABSTRACT

Deep learning and latest machine learning technology heralded an era of success in data analysis. Accompanied by the ever increasing performance, reaching super-human performance in many areas, is the requirement of amassing more and more data to train these models. Often ignored or underestimated, the big data curation is associated with the risk of privacy leakages. The proposed approach seeks to mitigate these privacy issues. In order to sanitize data from sensitive content, we propose to learn a privacy-preserving data representation by disentangling into public and private part, with the public part being shareable without privacy infringement. The proposed approach deals with the setting where the private features are not explicit, and is estimated though the course of learning. This is particularly appealing, when the notion of sensitive attribute is "fuzzy". We showcase feasibility in terms of classification of facial attributes and identity on the CelebA dataset. The results suggest that private component can be removed in the cases where the the downstream task is known a priori (i.e., "supervised"), and the case where it is not known a priori (i.e., "weakly-supervised").

## 1 INTRODUCTION

In recent years, learning with DNNs has brought impressive advances to the state-of-the-art across a wide variety of machine-learning tasks and applications. Yet, these approaches are generally only capable of significant performance leaps when large amounts of training data are provided for training purposes. However, building and curating such large data corpora comes with many strings attached. Firstly, it is cumbersome, expensive as well as time consuming to amass sufficient and foremost clean data. Apart from that, amassing large amounts of data increases the risk of privacy creep, i.e. subtly encoding privacy related information Narayanan & Shmatikov (2006); Backstrom et al. (2007). In this regard, many datasets have been released into the public domain that are unintentionally permeated with private information about individuals, raising serious concerns about data privacy. In Narayanan & Shmatikov (2006) anonymization could be reversed by making use of publicly available data, e.g. movie reviews. In Wachinger et al. (2015) showed, that the surface of magnet resonance images (MRI) of the brain can be used to identify people potentially allow for unintended disease progression. Growing privacy concerns will entails the risk of becoming a major deterrent in the widespread adoption of machine learning and the attainment of their concomitant benefits. Therefore, reliable and accurate privacy-preserving methodologies are needed, which is why the topic lately has enjoyed increased attention in the research community.

Several efforts have been made in machine learning to develop algorithms that preserve user privacy while achieving reasonable predictive power. This is even more challenging in the client-server scenario. Therein, the clients are supposed to send information to the server, which in turn then performs operations such as training for the client. The crucial aspect in this case is about the client's confidential data. As an example, consider a set of clients that aim at collaboratively learning an attribute classifier on face images, while preserving the identities of the individuals. Ideally the trained model, which classifies non-sensitive attributes (e.g., having glasses or not) with high accuracy, at the same time fails in classifying sensitive attributes of them (e.g., gender).

A standard approach to address the privacy issue in the client-server setup is to anonymize the *data* of clients. This is often achieved by directly obfuscating the private part(s) of the data and/or adding random noise to raw data. Consequently, it is the noise level controlling the trade-off between

predictive quality and user privacy (e.g., Differential Privacy Dwork (2006); Abadi et al. (2016)). These approaches, associate a privacy budget with all operations on the dataset. Complex training procedures run the risk of exhausting the budget before convergence.

Another widely adopted solution is rely on encoded data *representation*. Following this notion, instead of the client's data a feature representation is transferred to the server instead. Unfortunately, the extracted features may still contain rich information, which can breach user privacy Osia et al. (2017; 2018). Specifically, in the example above, an attacker can exploit the eavesdropped features to reconstruct the raw image, and hence the person on the raw image can be re-identified from the reconstructed image Mahendran & Vedaldi (2015).In addition, the extracted features can also be exploited by an attacker to infer private attributes Salem et al. (2019).

A recent solution to such a problem has been federated learning McMahan et al. (2016); Geyer et al. (2017), which allows us to collaboratively train a centralized model while keeping the training data decentralized. The idea behind this strategy is that the clients transfer the *training gradients* of data to the server instead of the data itself. While such an approach is appealing to train a neural network with data hosted in different clients, it does not allow the use of a centralized model for making a prediction at test time. Furthermore, transferring the models between clients and server entails significant data transmission, which considerably prolongs training. What is more, averaging the gradients across the clients further slows the backpropagation.

Most importantly, all of the aforementioned methods are mainly applicable when the private attributes are known a priori. That his, these approaches typically fail to prevent privacy attacks when the private information contained in the dataset is not explicitly identified. Some scenarios also defy simple annotation of private content, e.g. imagery of military and civilian commodities. In such scenarios, it is highly desirable to automatically remove content that may be subject to sensitive information. This situation is further aggravated in domains such as low-shot learning with scarcity of training examples and associated privacy labels, entailing ambiguity w.r.t. sensitive features (i.e., sensitive features are the ones reveal privacy).

In this paper, we focus on the following fundamental questions: how can we learn representations of a dataset in order to minimize the amount of information which could be revealed about the identity of each client? The main goal is to enable an analyst to learn relevant properties (e.g. regular labels non-privacy infringing) of a dataset as a whole, while protecting the privacy of the individual contributors (private labels, which can identify a client). This assumes the database is held by a trusted server that can learn a privacy-preserving representations, i.e. by sanitizing the identity-related information from the latent representation. Specifically, we postulate the decomposition of the latent representation into two latent factors: *style* and *content*. In this regard, style captures the private aspects of the data, whereas content encodes the public part. Thus, it is the public part that should be used for training downstream tasks as it can be transferred without compromising privacy. Following this notion, style encodes patterns that are shared among the samples of each client. In contrast to that, the content encodes information about concepts, which is shared across clients. Ultimately, this implies a disentanglement in the feature space in. *private features* and *public features*.

Our proposed method is built on top of the popular Variational Autoencoders (VAE) Kingma & Welling (2013), where it used as core representation learning paradigm. VAE consists of two networks, namely an encoder and a decoder. The former maps a data sample to a latent representation, while the latter maps this representation back to data space. VAE networks are trained by minimizing a cost function that encourages learning a latent representation, which leads to realistic data synthesis while simultaneously ensuring sufficient diversity in the synthesized data. This is achieved by minimizing the distance between input and reconstruction subject to distributional regularization on the latent space. In addition, to the VAE entailed cost functions, we augment the loss space with additional terms namely "content classification loss" and "style confusion loss". In context of a supervised setup, the former is utilized to enforce for target predictability for a downstream task, while the latter encourage preserving the privacy. We show that, this two additional terms can enforce disentanglement of the private and public parts of the representation, as well utilization in context of weakly supervised scenario, where the downstream attributes are not known a priori.

In summary, the main contributions of this paper are the three-fold. **First**, we pose the privacy-preserving representation learning problem as learning disentangled representations in a client-

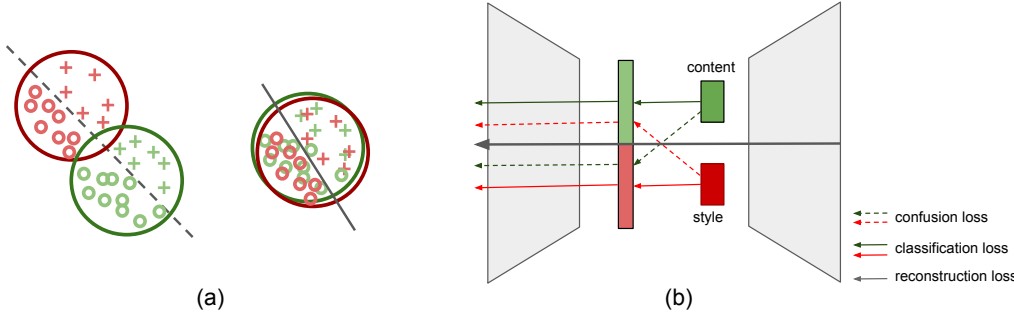

Figure 1: **Our proposed method:** **(a)** Schematic illustration of the general idea: Here we have two private information attributes (red and blue) and two public information attributes (circle and cross). The dashed line is a classifier trained in public information only. Our method, in solid line, both learns to classify the public information and preserve private information. **(b)** Our proposed architecture: the architecture is based on three sub-networks: one going from input to representation, one from representation to predicted regular labels, and one from representation to predicted private labels.

server setup. **Second**, we propose to learn disentangled representations from the client-level supervision by adding two novel loss terms to VAE. **Third**, we demonstrate experimentally that our proposed method learns a semantically meaningful privacy-preserving disentangled representation of client data.

## 2  RELATED WORKS

Our method is different from Dwork et al. (2017); Dwork (2006); Ryoo et al. (2017); Abadi et al. (2016), which focus on learning privacy-preserving models by perturbing raw data before sending them to the server. A key difference compared to these methods is that they share the data directly to the server, whereas our method proposes to send representation instead. The advantages of a representation sharing for client-server learning have been investigated recently in Osia et al. (2017; 2018). Nevertheless, such representation is proven to contain some privacy revealing information of clients. The recent success of adversarial learning has led to the increased adoption of this technique for learning representation to preserve sensitive information in different type of data. For instance, Srivastava et al. (2019) proposed to learn privacy-preserving representations for automatic speech recognition (ASR). In Yang et al. (2018) a representation is learned on the raw student clickstream event data, captured as they watch lecture videos in massive open online courses. In Li et al. (2019), the authors proposed an obfuscator that is designed to hide privacy related sensitive information from the features using adversarial training. Similarly, Kim et al. (2019) is based on adversarial learning, which encodes images to obfuscate the patient identity while preserving enough information for a medical segmentation task. Pittaluga et al. (2019) considered a formulation based on adversarial optimization between the encoding function and estimators for the private tasks. In our work, however, a completely different perspective on representation sharing is selected. We specifically propose a representation learning method, which facilitates disentanglement of private and public features at minimal additional training cost. This leads to a more privacy-preserving representation, while maintaining the method complexity at manageable level.

Most related to our paper is the work by Feutry et al. (2018), which aims at learning representations that preserve the relevant part of the information while dismissing information about the private labels corresponding to the clients' identity. A key difference compared to this method is that they require labels for the downstream task during representation learning. In contrast to that, our setting allows for testing on unseen downstream tasks - see Sec. 3.2. Very recently, Chen et al. (2018) proposed a complex method for privacy-preserving representation learning. However, to our knowledge, this is the first work proposing to disentangle private and public features using only the client level information.

At last, we note that our model is different from the federated learning methodology in McMahan et al. (2016); Geyer et al. (2017), which focuses on learning a decentralized private model by sharing gradient updates instead of representations. While we do not consider their framework here, our method could also be extended to those settings, leading to interesting future research directions. Other related works include Bouchacourt et al. (2018); Ganin & Lempitsky (2014); Creager et al. (2019)

## 3 PROPOSED METHOD

We want to learn representations that, beyond being private for a variety of sensitive attributes, can be adapted simply and compositionally for predicting many test-time task labels. We consider a two-stage client-server learning, where in the first stage we learn a private representation of the data, and in the second stage, the actual downstream task is performed. Our approach to this problem involves learning a disentangled representation that allows for easy sensitization of latent representation in terms of privacy. Specifically, we isolate information about sensitive and non-sensitive attribute to separate subspaces, while ensuring that the latent space factorizes these subspaces independently. Naturally, depending on the assumptions related to the downstream task condition (if it is known a priori or not), we propose multiple variants. We consider the cases where the the downstream task is known a priori (i.e., supervised disentanglement), and the case where it is not know a priori (i.e., weakly-supervised disentanglement).

Ultimately, the goal is to learn a representation $z \in \mathbb{R}^m$ of data $x \in \mathbb{R}^k$ with $z = f(x)$ and $m << k$, which decomposes into two parts $(z_\star \in \mathbb{R}^m, z_\bullet \in \mathbb{R}^n) = z$, where without loss of generality we assume $m = n$, representing the public and private information, respectively. Thereby, the public component should reveal as little as possible about sensitive information. That is we want to learn a function $f$ that learns the representation s.t. $z_\star \perp z_\bullet$. Ideally, this representation has strong utility for a multitude of downstream tasks. To this end, we employ a Variational Auto-Encoder (VAE) Kingma & Welling (2013) in combination with regularization terms. The VAE serves the purpose that the input is compressed in a meaningful fashion. However, VAE alone does not consider learning a representation that is useful for downstream tasks or puts restrictions in terms of privacy of attributes or latent factors. Attaching additional constraints on the latent representation is non-trivial. This becomes even more challenging, if the downstream task is not known a priori Locatello et al. (2019).

The proposed approach makes use of a vanilla VAE architecture that assumes isotropic Gaussian as latent prior $p(z) = \mathcal{N}(0, I)$. Optimization of VAE entails maximization of the Evidence Lower Bound (ELBO) criterion,

$$L_{VAE}(p, q) = \mathrm{E}_{q(z|x)}[\log p(x|z)] - D_{KL}[q(x|z) \| p(z)],  \tag{1}$$

where the first time is the reconstruction loss, and the second one the Kullback-Leibler divergence w.r.t. the prior distribution. Typically, the associated encoder $q(z|x)$ and decoder $q(x|z)$ functions are realized with Gaussians, whose parameters $\theta_p, \theta_q$ are estimated using neural networks.

In order to facilitate disentanglement of the private and public component we utilize different multiple classification tasks. To this end we define a mapping based on a deep feed-forward architecture that takes as input the latent representation $z$ and predicts a label $y \in Y$, defined as $y = g(z, \theta)$, which is parameterized by $\theta$. Specifically, we distinguish between public and private attributes $Y_\star, Y_\bullet$, respectively. The first objective is to predict the public and private labels based on their section of the latent representation. That is,

$$\tilde{y}_\star = g_\star(z_\star; \theta_\star) \quad \text{and} \quad \tilde{y}_\bullet = g_\bullet(z_\bullet; \theta_\bullet),  \tag{2}$$

with the associated loss terms $L_{\theta_\star}(y_\star, \tilde{y}_\star)$, $L_{\theta_\bullet}(y_\bullet, \tilde{y}_\bullet)$, which are typically realized with cross-entropy.

An obvious solution is to just optimize the functional comprising the VAE together the classification terms, pushing for classification based on each latent sub-vector, yielding:

$$E(\theta_p, \theta_q, \theta_\star, \theta_\bullet) = \lambda_{VAE} \cdot L_{VAE} + \lambda_\star \cdot L_{\theta_\star} + \lambda_\bullet \cdot L_{\theta_\bullet},  \tag{3}$$

where $\lambda. \in \mathbb{R}$ represents a scaling parameter. However, following this notions leads to a presentation that does not fulfill $z_\star \perp z_\bullet$. That is, there is information leakage between the terms, particularly when the attributes to be classified are not strictly semantically disentangled. Generally speaking, the disentangled latent code should capture no more than one semantically meaningful factors of variation in the data w.r.t. private and public attributes, respectively. This is in particular problematic, when there is excess information capacity in the bottleneck layer generating the latent representation.

Thus, in order to push for disentanglement, we employ adding label confusion terms. This entails adding, additional classification terms. The underlying notion is that sensitive attributes should not be predictable from public ones. The follow-up step, depends on the boundary conditions for learning the downstream tasks. Therefore, we decompose the into two variants. The first variant, assumes that the downstream task is a priori know. This allows, for an optimal disentanglement of public and private information, such that sensitive content is not compromised. The specifics of this scenario are discussed in Sec. 3.1. On the other hand, if the downstream task is not a priori know, the objective is to learn a representation that does not contain sensitive information in the public part, and yet is rich in terms of expressive power for a multitude of tasks. This is discussed in Sec. 3.2.

### 3.1 SUPERVISED DISENTANGLEMENT: LEARNING WITH TASK AND CLIENT INFORMATION

Consequently, we define the domain confusion classifiers according to,

$$\tilde{y}_{\neg \star} = g_{\neg \star} \left( z_\bullet ; \theta_{\neg \star} \right) \tag{4}$$

$$\tilde{y}_{\neg \bullet} = g_{\neg \bullet} \left( z_\star ; \theta_{\neg \bullet} \right), \tag{5}$$

which essentially assesses how well the private attribute can be derived from public latent part $z_\star$ and vice versa. As we, want to punish classifiability in the confused sense, we backpropagate in the negative gradient direction w.r.t. $\theta_{\neg \star}$ and $\theta_{\neg \bullet}$. This yields the following,

$$E \left( \theta_p, \theta_q, \theta_\star, \theta_\bullet \right) = \lambda_{VAE} \cdot L_{VAE} + \sum_i^2 \lambda_i \cdot L_i - \sum_j^2 \lambda_j \cdot L_j, \tag{6}$$

where we simplified the terms for the sake of economy in notation. The first sum captures from Eq. 2, whereas the second one incorporates the confusion classifiers from Eq. 5. In case the information between public and private label are not perfectly uncorrelated, optimization of the Eq. 6 allows for trading-off privacy vs. performance by adjusting $\lambda.$ parameters, accordingly.

### 3.2 WEAKLY-SUPERVISED DISENTANGLEMENT: LEARNING WITH CLIENT INFORMATION

In this scenario, the downstream task is not known a priori. Only, the client level information is given. Therefore, we propose to perform a counter-directional optimization. That is, while promoting the classifiability of the private content in the private latent variable, we penalize for the classifiability in the public latent variable. This boils down, to adding a loss term w.r.t. following classification objective,

$$\tilde{y}_{\neg \bullet} = g_{\neg \bullet} \left( z_\star ; \theta_{\neg \bullet} \right), \tag{7}$$

to the VAE loss of Eq. 1. Taken together, we yield following functional

$$E \left( \theta_p, \theta_q, \theta_\bullet \right) = \lambda_{VAE} \cdot L_{VAE} + \lambda_\bullet \cdot L_{\theta_\bullet} - \lambda_{\neg \bullet} \cdot L_{\theta_{\neg \bullet}}, \tag{8}$$

which maximizes the information w.r.t. the sensitive label in the private latent part, and minimizes the information in the public latent part. This constraint is considerably weaker compared to Eq. 5. However, it provides a general representation in absence of knowledge about future tasks. Furthermore, it assumes that the private and public are perfectly disentangled.

### 3.3 OPTIMIZATION

Optimization of the joint cost function Eq. 6 and Eq. 8 is non-trivial. On the one hand, the confusion terms have a strong destructiveness. Giving too much emphasis on confusion bears the risk of eliminating all information and maximizing entropy. On the hand, too much suppression of confusion

undermines the privacy aspects and reduces to the naive solution e.g. to Eq. 3. To this end, we propose to employ an annealing scheme. Specifically, in order to align the objectives, we perform pre-training without confusion. This allows, to learn a stable representation that allows for gentle modification, i.e. channeling of the information flow. Subsequently, we lower the scale factors $\lambda_\star, \lambda_\bullet$, while at the same time increasing $\lambda_{\neg\star}, \lambda_{\neg\bullet}$. Tapering off the classification objective scales w.r.t. the confusion avoids oscillations and facilitates stable learning behaviour.

## 4 EXPERIMENTS

We experimentally validate the proposed method using the MS-Celeb-1M dataset of celebrity face images Guo et al. (2016). This dataset contains a large number of identities (people) with multiple observations of each. The "in-the-wild" nature of the celebrity face images offers a richer testbed for our method as both identities and contingent factors are significant sources of variation.

**MS-Celeb-1M dataset.** From the aligned face images in the MS-Celeb-1M dataset, we select all celebrities that are represented by at least 8 images. Individual images are resized to $128 \times 128$ pixels. We split the celebrities into subsets of $80\%$ (training), $10\%$ (validation) and $10\%$ (test). The split is created in such a way that all celebrities are present in all sets. This leads to dataset containing images from $8732$ different actors and $40$ facial attributes. We scale the pixel values to $[1, 1]$, performing no additional pre-processing or data augmentation.

**Implementation details.** In order to learn the architecture, we employ Stochastic Gradient Descent (SGD) with momentum $0.9$ and weight-decay of $1e-4$ and a batch size of $96$. The learning rate ranges between $[0.0001, 0.1]$.

### 4.1 EVALUATION OF PRIVACY AND DOWNSTREAM TASKS

We validate two variations of our proposed method:

**Supervised Disentanglement.** In this experiments, we perform learning of the private representation and the downstream task jointly, assuming the downsteam task is known a priori. To do so, the model is at first trained without confusion loss term enabled. Upon convergence, the confusion loss is enabled, pushing for disentanglement between public and private attributes, respectively.

**Weakly-Supervised Disentanglement.** In this experiments, we first learn of the private representation only using the client-level supervision, and later employ a classifier the downstream task, simulating the case where the downsteam task is unknown a priori. Similarly, to the supervised case, the model is initially trained without confusion loss. Upon convergence, the confusion loss is enabled to push the disentanglement w.r.t. the private attribute only.

In this regard, two aspects are evaluated. On the one hand, the classification accuracy of public and private attributes on their associated public and private section of the learned latent representation. On the other hand, the domain confusion classifier accuracy is assessed as a degree of disentanglement. Lower accuracy in terms of domain confusion accuracy that is predictability of private labels from public latent space and vice versa, is an indicator for disentanglement.

As can be seen in Tab. 4.1, in the supervised setup explain in Sec. 3.1 demonstrates best performance in almost all categories, that is high classification accuracy and low accuracy for domain confusion.

| Method | ID | Att. | ¬ID | ¬Att. |
|---|---|---|---|---|
| VAE (Kingma & Welling (2013)) | 0.25 | 0.881 | - | - |
| ML-VAE (Bouchacourt et al. (2018)) | 0.155 | 0.884 | 0.153 | 0.882 |
| Our Method (*Weakly-Supervised*) | 0.756 | 0.521 | **0.0** | 0.909 |
| Our Method (*Supervised*) | **0.770** | **0.913** | 0.001 | **0.202** |

Table 1: Performance assessment for different methods on CelebA on the test split.

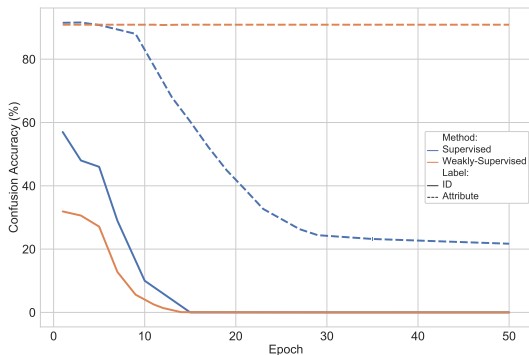

Figure 2: Confusion accuracy of different approaches. Blue indicates supervised approach, red semi-supervised. Solid line is the accuracy on ID classification, dashed average accuracy on attributes.

## 4.2 ABLATION STUDY ON COMPONENTS OF THE METHOD

In order to assess the contribution of each component, we assessed the performance of each module separately, gradually adding components for both supervised (see Tab. 4.2) and weakly supervised scenario (see Tab. 4.2), respectively.

First, the results indicate that learning a classifier directly on the VAE representation works to a certain degree on ID, and well on public attributes. Next, adding the classification term for both supervised and weakly-supervised scenario boosts the attribute and ID accuracy as expected. Simultaneously, high confusion accuracies in this configuration give clearly rise to missing disentanglement between the components. Subsequent adding the confusion team leads to a significant and desirable drop in accuracy w.r.t. confused classifier. However, whereas this significant drop appears to both public and private attributes in the supervised case, it only applies to the private attribute in the weakly-supervised case as expected. At the same time, classification accuracy on public attributes is barely affected by the confusion terms. What is more, although the classification accuracy drops on the the public (a priori unknown) attributes, it still performs at reasonable accuracy. See Fig. 2 for an illustration of the change behaviour in accuracy w.r.t. confused classifier.

| Method | ID | Att. | ¬ID | ¬Att. |
|---|---|---|---|---|
| Ours (only reconstruction loss) | 0.25 | 0.881 | - | - |
| Ours (rec. loss + classification loss) | 0.696 | **0.916** | 0.538 | 0.914 |
| Ours (rec. loss + cls. loss + confusion loss) | **0.770** | 0.913 | **0.001** | **0.202** |

Table 2: Ablation analysis for our methods on CelebA on the test split in the *supervised scenario*.

| Method | ID | Att. | ¬ID | ¬Att. |
|---|---|---|---|---|
| Ours (only reconstruction loss) | 0.25 | **0.881** | - | - |
| Ours (rec. loss + classification loss) | 0.752 | 0.555 | 0.311 | 0.909 |
| Ours (rec. loss + cls. loss + confusion loss) | **0.756** | 0.521 | **0.0** | 0.909 |

Table 3: Ablation analysis for our methods on CelebA on the test split in the *weakly-supervised scenario*.

## 5 CONCLUSION

The approach presented in this paper proposes an algorithm that yields a data representation that facilitates disentanglement into public and private components. The underlying notion is to make the public part shareable without privacy limitations. Subsequent, classification experiments confirm particularly strong performance in the fully-supervised scenario, where public attributes are

known beforehand. What is more, classification utility of public shareable representation is also demonstrated in the challenging privacy aware scenario, when the downstream task is not known a priori. At the same time, results suggest that disentanglement is successful between the components. Future work will entail to further close the performance gap between the these the fully supervised and the privacy aware scenarios.

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
