# OpenReview forum: "Privacy-preserving Representation Learning by Disentanglement"
_ICLR.cc/2020/Conference — Reject_

### Official Review · AnonReviewer2 · 2019-10-13
**Official Blind Review #2**

**Rating:** 1

**Review:**

PRIVACY-PRESERVING REPRESENTATION LEARNING BY DISENTANGLEMENT

Summary
This paper introduces a method to disentanglement the private and public attribute information in representation learning.

Strength:
1. The idea of introducing the confusion term to disentanglement private and public information seems novel.
2. The problem studied in this paper is very important.

Comments:
1. The existing results are not sufficient to validate the effectiveness of the method to prevent privacy leakage. More comparison with other previous methods should be conducted. For example, the previous work [1] has both theoretically and experimentally validated the effectiveness of DP based methods for preventing attribute attack. Recent work[2] also tries to reduce information leakage in representation learning.
2. Some important related works are missing. The difference between the proposed method and previous works with the same purpose should be made more clearly. See comment1 for some concrete examples of previous works.
3. The notation of $z$ is confusing. The author denotes $z=(z_{*}, z_{.})$, however, these three vectors' dimensions are the same, which really confuses me.
4. Some details of $L_{VAE}$ are missing. According to Equation~1., the author uses $z$ to build the reconstruct loss. However, based on previous notations, $z$ comprises of both private and public representations. So, what's the strategy to combine these two representations (concat?)?
5. Typos. For example, blue-> read in the caption of Figure 1.
6. The threat model in this paper needs to be made more clearly. The method proposed in this paper can only be used in the context where the private attribute is well-defined. There are many other threat models in secure machine learning research, such as membership attack and adversarial attack, which are not covered by this paper (My suggestion is adding a specific sub-section of threat model and do not use  "privacy-preserving" in the original title ).
[1] Privacy Risk in Machine Learning: Analyzing the Connection to Overﬁtting
[2] Mitigating Information Leakage in Image Representations: A Maximum Entropy Approach

**Experience Assessment:**

I have published one or two papers in this area.

**Review Assessment: Checking Correctness Of Derivations And Theory:**

I carefully checked the derivations and theory.

**Review Assessment: Checking Correctness Of Experiments:**

I carefully checked the experiments.

**Review Assessment: Thoroughness In Paper Reading:**

I read the paper at least twice and used my best judgement in assessing the paper.

---

### Official Review · AnonReviewer3 · 2019-10-22
**Official Blind Review #3**

**Rating:** 3

**Review:**

This paper brings to our attention the problem of privacy-preserving representation learning.
The assumption in this work is that the data feature can be split into two parts: private and non private.

The authors propose to use VAE with additional regularizer terms to learn hidden representations that would encode both private and non-private part as independent as possible.
There are two scenarios considered:
1) supervised disentanglement, total of 4 terms (two for public part, two for private part) are added to the VAE loss.
2) weakly-supervised disentanglement: in this setting, downstream task is unknown and only two terms are added tot he VAE loss: one to penalize the classifiability using private information in the public z space, and other to promote the classifiably using private information in the private z space.

Overall, I think it is a well-stated problem and this work seems to get some positive results on CelebA dataset. However, a few questions must be clearly addressed:
1. Eqn 6 is derived from Eqn 2 and Eqn 5. please clean up the notation on L terms. Currently they are not consistent and L_i and L_j are not defined.

2. In section 3.3, the authors mention the optimization is nontrivial. Can you expand on that? current section 3.3 is too short for the readers to appreciate that non-triviality.

Other comments:
On Page 1,
growing privacy concerns will entails —> entail
the "Glass / gender" example doesn’t make sense to me. why glass is considered non-private while gender is private? maybe we should use another example here.

On Page 2,
 penultimate paragraph, where it used as —> where it is used as …
While the latter encourage preserving —> encourages
This two additional terms —> these two additional terms

Page 7, Figure 2 caption needs rewriting.




**Experience Assessment:**

I do not know much about this area.

**Review Assessment: Checking Correctness Of Derivations And Theory:**

I assessed the sensibility of the derivations and theory.

**Review Assessment: Checking Correctness Of Experiments:**

I carefully checked the experiments.

**Review Assessment: Thoroughness In Paper Reading:**

I read the paper at least twice and used my best judgement in assessing the paper.

---

### Official Review · AnonReviewer1 · 2019-10-27
**Official Blind Review #1**

**Rating:** 1

**Review:**

This paper investigates the use of variational auto-encoders (VAEs) and disentanglement to create high quality data representations that hide sensitive attributes. The authors consider two settings: (1) a supervised setting where both the sensitive labels and the downstream machine learning task labels are available to the data holder, and (2) a weakly supervised setting where only sensitive attributes are available. For both settings, the authors propose creating data representations that have 2 components: component 1 captures any information in the data bout the sensitive attributes, and component 2 captures everything else (especially the information useful for a downstream machine learning task). The goal is to ensure that these two components are disentangled.  This is done by training classifiers on each component: 2 classifiers that try to reconstruct the sensitive attributes from each component separately (and in the supervised setting, 2 other classifiers that guess the downstream ML task from each component separately). The overall loss captures all components and ensures that one cannot reconstruct the sensitive attributes from component 2.

Overall, the paper addresses an interesting and timely problem that is of great relevance to this community. However, despite being generally clear, the paper has many grammatical errors and typos. In its current shape, the paper cannot be published.

More importantly, the paper has a number of issues that need to be improved:

1. The introduction makes a number of claims that are incorrect. For instance, the introduction claims that under differential privacy (and refers to Abadi et al.), machine learning models are learned from anonymized data and that complex training procedures run the risk of exhausting the budget before convergence. This is not true because under the classical setting of differential privacy, the models are trained on clean data BUT the process of learning is anonymized (see DP-SGD from Abadi et al.). More importantly, recent research and results in this space indicate that one can train high quality machine learning models with strong differential privacy guarantees (check McMahan et al. ICLR18: Learning Differentially Private Recurrent Language Models). Further, the authors claim that Federated Learning does not allow for the use of the trained model in a central setting. This is also not true because federated learning is an approach to train models on massively decentralized data -- in a way that is orchestrated by a service provider. The learned model can be used in a centralized or decentralized service. Moreover, the communication cost of training models under federated learning may be lower than communicating the data (several hundreds if not thousands of high resolution images) to a central service provider, so the claims about communication are not always correct. Finally, the authors claim that federated learning and differential privacy are useful when the private attributes are known a priori and these approaches fail to provide privacy protections when the private information contained in the dataset isn't identified. This is wrong. Both approaches do not require the knowledge of private information. They, however, require the knowledge of the downstream machine learning task -- something that may not always defined a priori.

2. The privacy guarantees provided by the authors' approach are rather weak and unclear. (1) They assume that a trusted data holder already access to the dataset and wants to release private representations. In the supervised setting, why not just revealing the learned model? In the weakly supervised setting, where will the downstream machine labels come from? Are these representations released only for unsupervised learning tasks? This should be clarified -- and the experiments at the end should reflect this. (2) The privacy guarantees are with respect to (a) a computationally and statistically bounded adversary (i.e. there may very well be stronger adversaries with access to side information that can perfectly reconstruct the sensitive attributes),  and (b) pre-defined sensitive attributes (i.e. there may be other sensitive attributes that one can learn from the published representations that are not captured by this framework).

3. The paper makes no attempt to properly survey the literature on learning representations under censorship and fairness constraints. For example, they do not reference and compare against: (a) Censoring Representations with an Adversary (https://arxiv.org/abs/1511.05897), (b) Learning Adversarially Fair and Transferable Representations (https://arxiv.org/abs/1802.06309), (c) Context-Aware Generative Adversarial Privacy (https://arxiv.org/abs/1710.09549), (d) Learning Generative Adversarial RePresentations (GAP) under Fairness and Censoring Constraints (https://arxiv.org/abs/1910.00411), and many others. Without a clear (empirical) comparison to these works, the benefits of the proposed approach are unclear.

4. The proposed method for disentanglement does not scale to non-binary sensitive attributes (because of the blowup in the number of classifier pairs that would need to be trained to ensure disentanglement). Thus, this approach may be limited to cases with a few sensitive attributes.

5. The authors are encouraged to show the learned representations (so that one could verify with the naked eye that the representations are indeed disentangling the sensitive attributes).


**Experience Assessment:**

I have published in this field for several years.

**Review Assessment: Checking Correctness Of Derivations And Theory:**

I carefully checked the derivations and theory.

**Review Assessment: Checking Correctness Of Experiments:**

I carefully checked the experiments.

**Review Assessment: Thoroughness In Paper Reading:**

I read the paper thoroughly.

---

### Public Comment · ~Jay_Pavagadhi1 · 2019-10-08

Strength:
- Excellent clarity in presentation & well-written paper.
- The idea behind paper is important, especially the learning of publicly shareable representation that doesn't compromise privacy.
- Also, I liked the fact that grouping samples by source can easily be used to learn private attributes and can boost overall accuracy. In many classification tasks, this is often overlooked feature of dataset.


Weakness:
- The author mentions about federated learning and claims that proposed methods in paper can be extended to those settings. However, if I am not mistaken, the basic premise of federated learning is to have small number of samples on very large number of devices and learning representation from those small samples may not be plausible.
- I wish author could have added some visualization of sample images and their reconstructions for VAE and proposed methods.
- Also, as desired, ID classification accuracy dropped from 77% to 0, I was wondering if model just swapped ID of one actor with other's creating sort of a pair. It would be really interesting to see, how many images of an actor misclassified to another specific actor.


Typos:
- where the first time is the reconstruction loss -> where the first term is the reconstruction loss
- the ever increasing performance -> the ever-increasing performance
- in the cases where the the downstream -> in the cases where the downstream
- this two additional terms -> these two additional terms
- where it is not know a priori -> where it is not known a priori
- In this experiments -> In this experiment
- accuracy drops on the the public -> accuracy drops on the public
- task is a priori know -> task is a priori known
- in the supervised setup explain in -> in the supervised setup explained in
- adding the confusion team leads to -> adding the confusion term leads to
- (red and blue) -  (red and green)
- That his, these approaches ->  That is, these approaches

---

> ### Author Response · Authors · 2019-10-11
> **Extension to federated setup; Visualization of public/private representation.**
>
> We thank the reviewer for the interest in our work and we appreciate the very insightful comments.
>
> i) Federated Learning: Learning disentangled representations in the federated setting is a challenging endeavor. That is why we pose it as a future research direction in the paper. Extending the proposed method to a federated setup can be achieved on different levels of decentralization.
> In the simplest case, the objective is to learn a downstream task in a decentralized fashion on feature representation. This scenario assumes that the disentangled feature representation is learned beforehand via a trusted curator. Then the downstream task is learned decentralized via “federated averaging” [1].
> The more challenging scenario entails not only learning the downstream task in a decentralized fashion but also the disentangled representation. The challenge arises here as the domain confusion classifier w.r.t. sensitive attributes has to be learned locally on each client. Consequently, as the client-level supervision (i.e., IDs) is not accessible in a federated optimization. One possible solution is to divide the model parameters into public and private parameters.
> Consequently, the public parameters can be trained in a decentralized fashion by federated averaging [1], while the private parameters have to be optimized locally on each client. Eventually, both the aggregated public and local private parameters are used for the downstream task. A similar strategy is proposed in the work of [2]. It should be noted that such a solution based on parameter splitting does not guarantee any sorts of semantic disentanglement between the public and the private parts of the representation. This contrasts with the underlying idea of the proposed approach, where disentanglement is achieved through employing the confusion loss at the client-level.
>
>
> ii) VAE reconstruction visualization: The VAE is used as a backbone for learning disentangled representation, with the goal of utilizing the representation for a downstream task. Therefore, the visual aspect of reconstruction itself is of minor importance. However, we agree that the visualizations provide further insights into the disentanglement behavior. Specifically, we noticed that visualizations from private latent and public parts confirm the disentanglement visually. More concretely, reconstructions from the private part of the representation result in a sharpening of identity revealing properties such as skin color, eye style. Simultaneously, these visualizations feature vanishing of public attributes such as sunglasses, jewelry. In contrast to that, reconstructions from the public representation part contain generic facial features, e.g., smiles, teeth, and face outlines. We plan to add visualizations in the final version of the manuscript.
>
> iii) ID classification: We analyzed the predictive pattern of the ID association. We did not observe any pattern related to, i.e., swapping pairs. We would add further detail on that in the final version of the manuscript to give more insight in this regard.
>
> We thank the reviewer for hinting at the typos. The final version of the manuscript will accommodate these along with suggestions made.
>
> [1] “Communication-Efficient Learning of Deep Networks from Decentralized Data”, McMahan et al., AISTAT 2016
>
> [2] “Federated User Representation Learning”, under review, ICLR 2020, https://openreview.net/forum?id=Syl-_aVtvH

---

### Decision · Program_Chairs · 2019-12-19

**Decision:**

Reject

**Comment:**

The paper leverages variational auto-encoders (VAEs) and disentanglement to generate data representations that hide sensitive attributes. The reviewers have identified several issues with the paper, including its false claims or statements about differential privacy, unclear privacy guarantee, and lack of related work discussion. The authors have not directly addressed these issues.